# Achieving environmental stability in an atomically thin quantum spin Hall insulator via graphene intercalation

Cedric Schmitt [1,2,9], Jonas Erhardt [1,2,9], Philipp Eck [2,3], Matthias Schmitt [1,4], Kyungchan Lee[1,2], Philipp Keßler [1,2], Tim Wagner [1,2], Merit Spring [1,2], Bing Liu [1,2], Stefan Enzner [2,3], Martin Kamp [1,5], Vedran Jovic[6,7], Chris Jozwiak [8], Aaron Bostwick [8], Eli Rotenberg [8], Timur Kim [4], Cephise Cacho [4], Tien-Lin Lee[4], Giorgio Sangiovanni [2,3], Simon Moser[1,2] & Ralph Claessen [1,2] ✉

Atomic monolayers on semiconductor surfaces represent an emerging class of functional quantum materials in the two-dimensional limit — ranging from superconductors and Mott insulators to ferroelectrics and quantum spin Hall insulators. Indenene, a triangular monolayer of indium with a gap of ~ 120 meV is a quantum spin Hall insulator whose micron-scale epitaxial growth on SiC(0001) makes it technologically relevant. However, its suitability for room-temperature spintronics is challenged by the instability of its topological character in air. It is imperative to develop a strategy to protect the topological nature of indenene during ex situ processing and device fabrication. Here we show that intercalation of indenene into epitaxial graphene provides effective protection from the oxidising environment, while preserving an intact topological character. Our approach opens a rich realm of ex situ experimental opportunities, priming monolayer quantum spin Hall insulators for realistic device fabrication and access to topologically protected edge channels.

With the promise of dissipation-less spin-polarized boundary modes, quantum spin Hall insulators (QSHI) could initiate a paradigm shift in future spintronics technologies. The conceptual application perspective is bright and ranges from spin-transistors[1,2], to low-power consumption devices[3,4], to QSHI-based quantum computing[5]. However, finding suitable materials for practicable device realization faces major challenges. The band-inverted narrow-gap semiconductors for which the quantum spin Hall effect had first been demonstrated[6,7] do not lend themselves to room-temperature (RT) applications. 2D Dirac semimetals formed by atomic honeycomb monolayers as motivated by the seminal work of Kane and Mele[8] are a promising alternative[9]. But while spin-orbit coupling (SOC) in graphene is too weak to open an appreciable band gap, monolayers built from heavier group IV elements such as silicene, germanene, and stanene[10,11] failed to display a large non-trivial band gap when placed on a semiconducting substrate[12].

In contrast, band-inverted large gap 2D Dirac semimetals have been successfully realized in group III and V monolayers on SiC(0001), specifically bismuthene[13] and the recently discovered indenene[14], and were experimentally confirmed as QSHIs. They could potentially solve

[1]Physikalisches Institut, Universität Würzburg, D-97074 Würzburg, Germany. [2]Würzburg-Dresden Cluster of Excellence ct.qmat, Universität Würzburg, D-97074 Würzburg, Germany. [3]Institut für Theoretische Physik und Astrophysik, Universität Würzburg, D-97074 Würzburg, Germany. [4]Diamond Light Source, Harwell Science and Innovation Campus, Didcot OX11 0DE, UK. [5]Physikalisches Institut and Röntgen Center for Complex Material Systems, D-97074 Würzburg, Germany. [6]Earth Resources and Materials, Institute of Geological and Nuclear Science, Lower Hutt 5010, New Zealand. [7]MacDiarmid Institute for Advanced Materials and Nanotechnology, Wellington 6012, New Zealand. [8]Advanced Light Source, Lawrence Berkeley National Laboratory, Berkeley, CA 94720, USA. [9]These authors contributed equally: Cedric Schmitt, Jonas Erhardt. ✉e-mail: claessen@physik.uni-wuerzburg.de

the device challenge, yet, are inherently unstable to environmental factors outside their ultra-high vacuum (UHV) birthplace. As a result, characterization has hitherto been bound to UHV-based surface science techniques such as angle-resolved photoemission (ARPES) and scanning tunneling microscopy (STM).

Here, we design an atomically thin protection to make these quantum materials operational in air, by placing quasi-freestanding graphene as protective sheet atop the QSHI monolayer via intercalation. Graphene's resilience to ambient conditions provides an efficient protection against oxidation, while it leaves the intercalated material unaffected as was shown for a variety of few-layer quantum materials[15,16]. With respect to topological physics, intercalation was suggested as a means to tailor the spin-orbit gap of graphene[17]. In contrast, here we reverse the roles of graphene and the intercalant, by using the former to stabilize the latter as a QSHI. For this purpose we employ indenene, the triangular monolayer phase of indium that can be grown routinely in high-quality monodomains on large areas of SiC[18]. It is particularly suited for wafer-sized intercalation, which not only protects both its structure and its topological character but also ensures its chemical integrity upon exposure to environmental conditions, as we demonstrate below.

## Results and discussion

In its pristine form, the topological electronic structure of indenene is the result of a synergetic interplay of the indium monolayer and its underlying SiC substrate. The latter breaks the surface mirror plane and gaps out the metallic indium $sp$ states, leaving a set of Dirac bands of in-plane $p$-orbital character located at the K-point of the Brillouin zone (Fig. 1a). The degeneracy of the Dirac point is lifted by two counteracting mechanisms[14,19]. While in-plane inversion symmetry breaking (ISB), induced by the topmost carbon atoms of the substrate (Fig. 1c), promotes a trivial band gap, atomic SOC drives the band

inversion and hence a large QSHI gap. As the bonding distance $d_{In-Si}$ between indium and the topmost Si-layer controls the ISB strength $\lambda_{ISB}$ felt by the In monolayer and consequently determines its topology and gap size, it has been used as one among several experimental indicators to indenene's topology. We thus find the pristine form to lie deep within the QSHI regime (Fig. 1c, blue data point)[14].

Intercalating indenene into the graphene/SiC interface, the massive Dirac bands are found to be well preserved, yet, significantly depopulated with $E_F$ shifted from n- to slightly p-type indenene (Fig. 1b). At the same time, the measured bonding distance $d_{In-Si}$ is consistent with that of pristine indenene, preserving SOC as the dominating factor, and stabilizing the topological gap along with the QSHI state (Fig. 1c, yellow data point), an assignment that we confirm by a second topological identifier further below.

The virtue of this graphene-covered QSHI relies on its resilience against oxidation, which we study by controlled oxygen exposure and subsequent X-ray photoelectron spectroscopy (XPS) on the pristine (Fig. 1d) and intercalated indenene (Fig. 1e). In both cases, the as-grown material (black spectra) reveals identical In $3d_{3/2}/d_{5/2}$ doublets. Exposing pristine indenene to large doses of oxygen (red) causes these peaks to broaden and display a chemical shift to higher binding energies, indicating strong indium oxidation[20]. In contrast, exposing intercalated indenene to the same dose of pure oxygen (red), ambient air (green) or even water (blue) has virtually no impact on the indium oxidation state nor on its band structure (see further below and Supplementary Note 5) and thus confirms the protective function of the graphene overlayer.

Having summarized the phenomenology, let us now focus on detailed aspects of indium intercalation[15,21,22], especially the large area growth of monolayer indenene and the identification of its non-trivial topology. Following the well-established recipes of metal intercalation, the synthesis is initiated by sublimating the topmost Si atoms off the SiC(0001) substrate under supporting Ar-atmosphere, leaving a C-rich

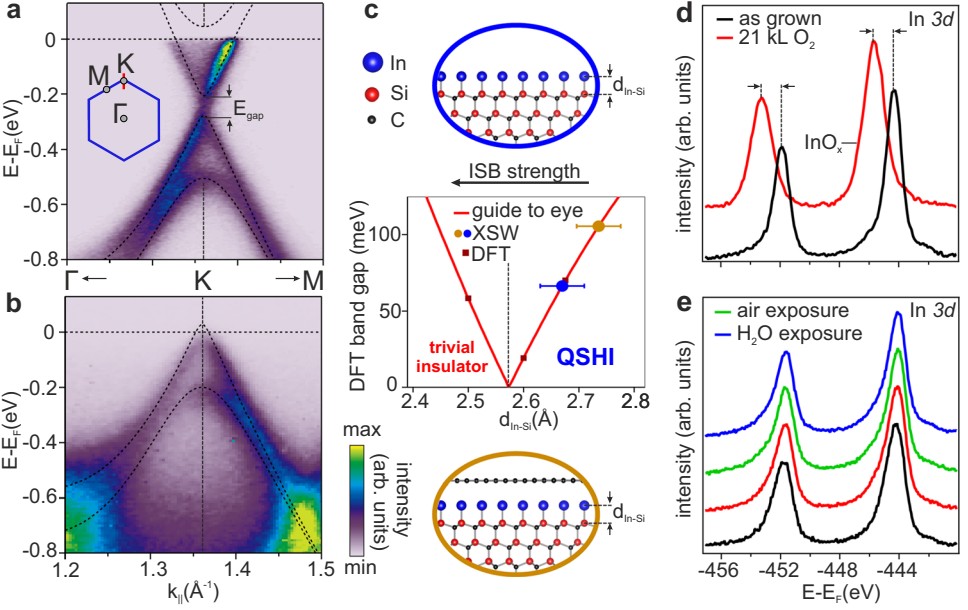

**Fig. 1 | Graphene-intercalated indenene is topologically non-trivial and resilient to atmosphere. a, b** ARPES spectra at the K-point of **a** pristine (20 K) and **b** intercalated indenene (RT) measured with 21.2 eV photons and overlaid DFT (HSE06) calculations (dotted lines) of pristine indenene. Spectra in **a,b** were extracted along the red path in the sketched indenene BZ (blue) and share the $k_\parallel$-axis. The band splitting in **a** and **b** originate from the combined role of SOC and ISB as discussed in the text. **c** DFT (HSE06) band gap calculations as well as a guide to the eye (red line) of pristine indenene as a function of the In-Si bond length $d_{In-Si}$, the latter controlling the ISB strength $\lambda_{ISB}$ as indicated by a black arrow.

Experimentally (by X-ray standing wave (XSW) photoemission) determined $d_{In-Si}$ of intercalated (yellow data point) and pristine (blue data point) indenene are placed in this diagram. **d, e** In $3d$ XPS core-level peaks of **d** pristine and **e** intercalated indenene, for the as-grown films (black; $E_{3/2} = -451.9$ eV; $E_{5/2} = -444.3$ eV), after exposure to 21 kL (red) of oxygen, after 10 min exposure to ambient air (green, only **e**), and after immersion in liquid water and subsequent mild *in vacuo* degas (blue) as specified in the Methods section. Panel **d, e** share the same energy-axis, $E_F$ being the Fermi energy. Black arrows and dotted lines in **d** indicate the chemically shift of oxidized In (InO$_x$).

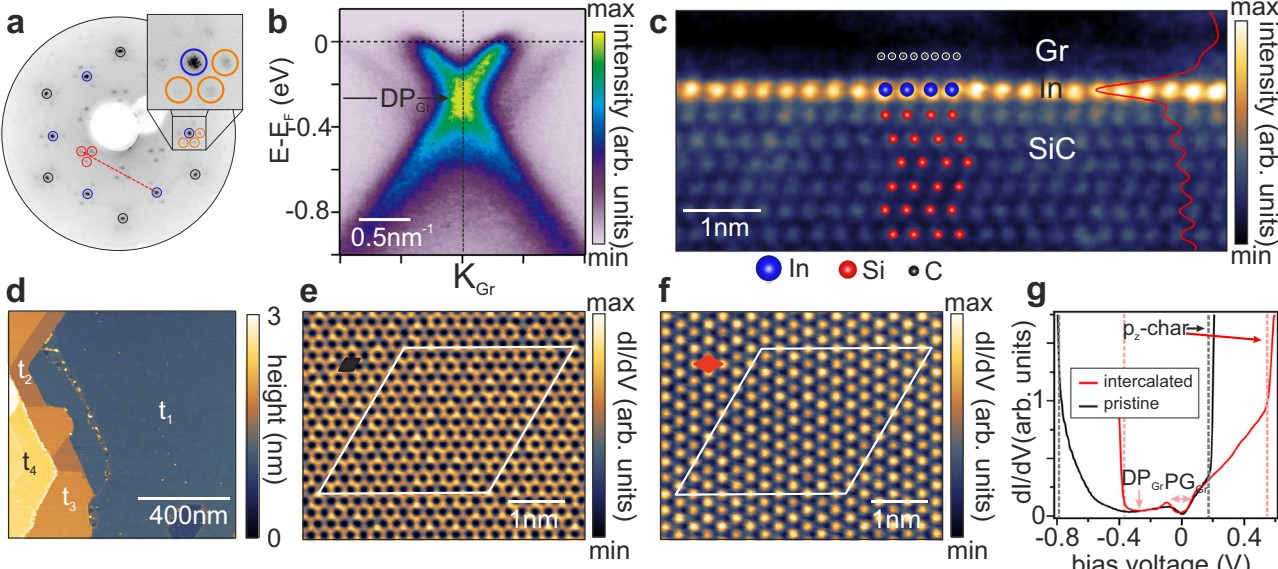

**Fig. 2 | Structure of intercalated indenene. a** LEED image taken at 100 eV showing diffraction spots of $(6\sqrt{3} \times 6\sqrt{3})$R30° periodicity (orange)[31], SiC(0001) (1 × 1) (blue) and graphene (black). Red marks indicate a selection of possible scattering vectors between SiC(0001) (1 × 1) and graphene[27]. **b** ARPES spectra taken at hv=46 eV around the graphene K-point $K_{Gr}$ indicate the graphene Dirac point ($DP_{Gr}$) position at ∼−0.22 eV. **c** RT STEM image of 1 ML indium intercalated graphene revealing positions of Si (of SiC, red spheres), indium (blue spheres) and graphene (black spheres), the latter being most evident in the horizontally integrated intensity profile (red). **d** STM constant current topography of intercalated indenene

(1 μm × 1 μm) measured at 4 V and 10 pA after immersion in water and a mild degas. Labels $t_{1-4}$ indicate different SiC terraces. **e, f** Lock-in dI/dV maps taken in constant height mode showing **e** the graphene lattice (black unit cell) at 50 mV and **f** the indenene lattice (red unit cell) at 800 mV. A white rhombus marks the $(6\sqrt{3} \times 6\sqrt{3})$ R30° moiré unit cell of both lattices in each panel. **g** dI/dV point spectroscopy of pristine (black) and intercalated indenene (red), both showing a sharp increase (dashed red/black lines) attributed to the onset of In $5p_z$-like states. Minima in the dI/dV spectroscopy of intercalated indenene are identified as the graphene $DP_{Gr}$ and an inelastic tunneling gap ($PG_{Gr}$)[25].

buffer layer referred to as zero-layer graphene[23,24]. In a cyclic process of indium deposition and subsequent annealing, we replace this carbon layer for indium as bonding partner to the substrate, hereby lifting the ZLG template from the subjacent SiC to form quasi-freestanding monolayer graphene (QFMG). The indium forms a bilayer below graphene right after the intercalation, the properties of which have already been reported[15] and are summarized in the Supplementary Note 1.

By thermal removal of In (550 °C, 30 min followed by 10 s at 800 °C) we now convert the indium bilayer into monolayer indenene, pushing it into the QSHI phase. This temperature is slightly higher than the intercalation temperature itself and thus provides the required energy to deintercalate indium as the inverse process to intercalation. The presence of QFMG after this treatment is evident from intense graphene spots in low energy electron diffraction (LEED; encircled black in Fig. 2a) and the characteristic π-band crossing of graphene in ARPES (Fig. 2b). Representative scanning transmission electron micrographs (STEM) (Fig. 2c) reveal the projected indenene layer, containing one In atom per Si site of the SiC surface. Large area indenene intercalation of graphene is confirmed by micron-sized STM scans revealing a uniform film, occasionally interrupted by steps of the SiC substrate (Fig. 2d).

Remarkably, the choice of the bias voltage selects which of both lattices can be probed at the atomic scale (Fig. 2e, f): Near the Fermi level, the hexagonal lattice of graphene appears (Fig. 2e), but increasing the bias voltage beyond 600 mV at the very same position, uncovers the 30°-rotated triangular (1 × 1) lattice of indenene (Fig. 2f). We attribute this to indenene $5p_z$ states that apparently have a larger overlap with the tip wavefunction than the smaller graphene $p$-states. Their onset is evident in a steep increase of the differential conductance marked by dotted lines in Fig. 2g and defining an energy window in which the graphene lattice is accessible. Within this window, we find the characteristic inelastic tunneling gap $PG_{GR}$[25] and a minimum in the dI/dV spectra readily identified as the Dirac point $DP_{Gr}$, see

Fig. 2b. The counterpart to this lattice-selective imaging is the orbital-selectivity in pristine indenene[14,19], where the in-plane $p_{x,y}$ Dirac states can only be accessed within the gap between the $p_z$ band edges. Their energy shift between pristine and intercalated indenene is in excellent agreement with the corresponding doping situation observed in ARPES spectra in Fig. 1.

To further elaborate on this link, we recapitulate the experimental band structure of the pristine 1 ML In film in Fig. 3a[18], and compare this to the intercalated counterpart in Fig. 3b. ARPES at the Γ-point of the pristine phase shows the intense SiC valence band maximum as well as an indium-related band dispersing upwards in energy, the latter exposing distinct maxima along the ΓM and ΓK paths (marked by blue arrows). These maxima arise from the substrate-induced breaking of mirror symmetry that fosters hybridization between in- and out-of-plane In $p$ orbitals[14]. In total, we recognize all ARPES features of the pristine phases to reappear in their intercalated counterpart. The latter additionally displays the intense graphene π-band and also faint replicas of both Dirac band structures (arrows at $E_F$ in Fig. 3b), caused by scattering of the outgoing photoelectron wave on the moiré lattice – an effect already known from other intercalated materials[26]. Similar multi-scattering between indenene and graphene lattices is also seen in our LEED data, see red marks in Fig. 2a and ref. 27.

With respect to pristine indenene, the intercalated indenene bands are shifted up by approximately 250 meV (horizontal lines in Fig. 3), corresponding to an overall charge carrier depletion of $\Delta n_{In} \approx -3 \times 10^{12}$ carriers per cm$^2$ that we estimate from the Fermi surface area (see Supplementary Note 4) and is in qualitative agreement with the n- to p-type transition in Fig. 1(a,b). At the same time, Fig. 2b suggests the graphene layer to accumulate $\Delta n_{Gr} \approx +3.4 \times 10^{12}$ cm$^{-2}$ with respect to charge neutral graphene, indicating a dominating electron transfer from indenene to graphene that is balanced by the underlying SiC substrate[28].

We conclude this ARPES comparison by noting that the data set of the intercalated indenene in Fig. 3b has been measured after an additional ex situ immersion of this particular sample into liquid water, followed by a mild anneal after reintroduction into the vacuum system. Remarkably, we find no changes in dispersion nor energy position of the indenene band structure before and after water treatment, demonstrating the enormous passivation strength of the single graphene overlayer (see Supplementary Note 5 for a detailed comparison).

Next we address the question, if intercalation has any effect on indenene's topology. Although we found the intrinsic indenene band dispersions to be unaffected by the graphene overlayer, the observed charge transfer indicates non-negligible interaction between indenene and graphene that could potentially affect both gap magnitude and its inverted character, for instance by modulating $\lambda_{ISB}$. To assess this effect, we first make the band gap fully accessible to ARPES by successive deposition of potassium that donates electrons to the heterostructure, thus populating the indenene conduction band step-by-step, see Fig. 4a, b and Supplementary Note 6. Indeed, increasing K-doping reveals a gap of $E_{gap} \approx 100$ meV in rigidly shifted indenene Dirac bands, consistent with the gap size found for pristine indenene[14].

Having established the persistence of the gap, we now turn to its topology. For this purpose, we first performed density functional theory (DFT) calculations for indenene without and with graphene cover as shown in Fig. 4c, d for the same In-Si bond length $d_{In-Si}$. The results clearly indicate that the bands at the valley momenta preserve their non-trivial sequence. The splitting due to the SOC strength $\lambda_{SOC}$ gets slightly renormalized but is roughly compensated by a concomitant reduction of the $\lambda_{ISB}$, confirming the effectively unchanged band gap observed in ARPES. Experimentally, using X-ray standing wave (XSW) photoemission, we find a bonding distance of intercalated indenene of $d_{In-Si} = (2.74 \pm 0.04)$ Å, which is consistent with pristine indenene and clearly situated in the QSHI sector of the phase diagram (cf. Fig. 1c, details in Supplementary Note 7).

This classification is corroborated by a second, more fundamental experimental indicator which is related to the orbital angular momentum ($L_z$) character of the gap-defining $p_\pm = p_x \pm ip_y$ Dirac states. As already demonstrated for pristine indenene[14], the band inversion of the topologically non-trivial phase is characterized by an alternating $L_z$ energy staggering. Via wavefunction interference this translates into an alternating charge localization in the unit cell, directly accessible to local density of states (LDOS) measurements by STM/STS.

Unfortunately, the gaphene cover complicates tunneling from these $p_\pm$ Dirac states, especially at energies with high graphene LDOS. We thus focus on bias voltages with small graphene LDOS near its Dirac point energy, where also the two topmost indenene valence band states VB-1 and VB are situated (Figs. 1 and 2). In order to disentangle the indenene and graphene contributions to the LDOS maps we filter the pronounced indenene (1×1) frequencies of the fast Fourier transform (FFT), that appear along with frequencies of the combined indenene-graphene ($6\sqrt{3} \times 6\sqrt{3}$)R30° moiré lattice (see Supplementary Note 8). Filtered d$I$/d$V$ maps taken at the energies of VB (−250 mV) and VB-1 (−100 mV) and at the exact same spatial position display a robust switch of the LDOS maximum from the right half of the unit cell for VB (site B in Fig. 4e) to the left half for VB-1 (site A in Fig. 4f), as expected exclusively in the QSHI phase. To ensure that we are indeed sensitive to VB-1 and VB of indenene, we reproduced this experiment for different doping situations and find the A-B switch to follow the valence bands on the energy axis (see Supplementary Note 8). This completes the experimental proof that graphene-intercalated indenene is topologically robust and can be clearly assigned to the QSHI regime.

Combined with its remarkable resilience against ambient conditions and even upon immersion in water, graphene intercalation of indenene opens up a wealth of experimental possibilities to characterize and manipulate this 2D topological insulator ex situ. While the conductive nature of the graphene cap may still interfere with meaningful edge transport measurements, its effective protection of the monolayer QSHI certainly paves the way for (nano)fabrication of device structures, e.g., for field effect gating, or for optical or infrared experiments such as Raman and Landau level spectroscopy in non-vacuum settings. As a perspective, hexagonal boron nitride (h-BN) could be an interesting alternative as large-gap inert capping layer, for which, however, efficient deposition and scalability pose serious challenges still to be met. In any case, our results clearly demonstrate that van der Waals capping is a viable route to bring otherwise highly delicate atomic monolayer systems into the realm of device physics while preserving their specific properties as ultimate 2D quantum materials, even in rough chemical environments[29].

## Methods
### Sample preparation
Pristine and intercalated indium films were prepared on atomically flat N-doped (0.01−0.03) Ωcm 4H-SiC (12 mm × 2.5 mm) substrates by a dry-etching technique that saturates the silicon dangling bonds with

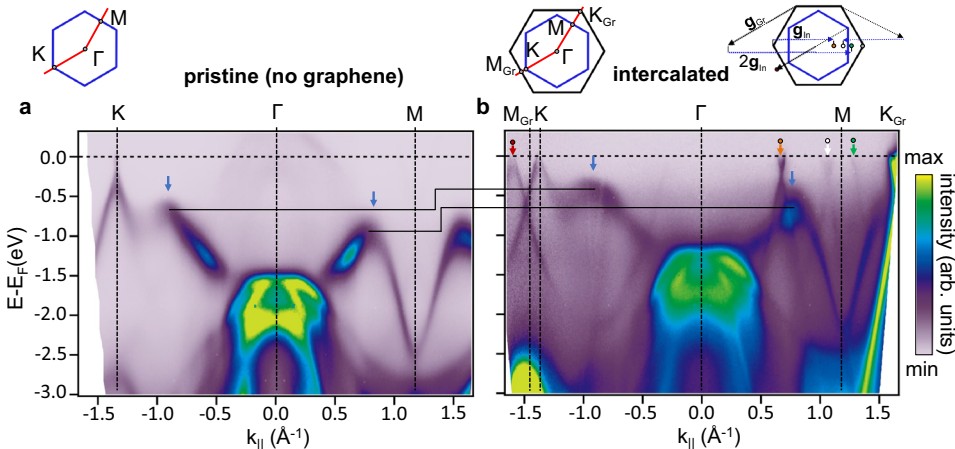

**Fig. 3 | The band structure of pristine and graphene capped indenene.** ARPES of **a** pristine monolayer indium and **b** intercalated indenene on SiC(0001). The data were taken at RT with $h\nu$=21.2 eV. Blue arrows indicate distinct band maxima due to out-of-plane mirror symmetry breaking and orbital hybridization. The different indenene band population is illustrated by black stepped line anchored at those maxima. The top row illustrations depict the Brillouin zones of indenene (blue) and graphene (black) and the high symmetry $k$-path (red) along which the ARPES data are shown. Graphene and indenene band replicas in **b** that are consistent with electron diffraction off the In/SiC (orange) or graphene lattice (red) and replicas consistent with multiple scattering (white, green) are shown in the sketch top right.

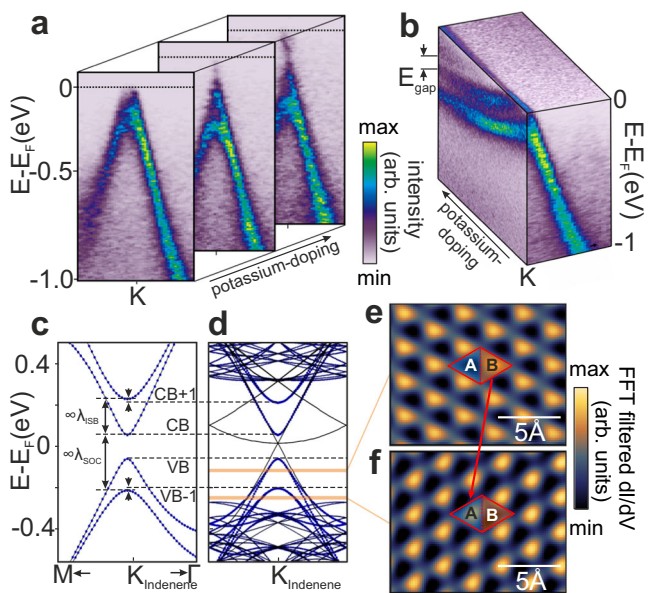

**Fig. 4 | QSHI character of intercalated indenene. a, b** Potassium doping of the indenene K-point in ARPES taken with $h\nu = 90$ eV photons. Increasing doping populates the conduction band, separated by an energy gap of $E_{gap} \approx 100$ meV from the rigidly shifted valence bands. **c, d** DFT (PBE) band structure (black) around the K-point of pristine **c** and intercalated indenene **d**. The blue marker radius denotes the indium character of the bands. Band splitting among VBs (CBs) is driven by the ISB strength $\lambda_{ISB}$ (arrow), while the larger $\lambda_{SOC}$ splitting (arrow) opens a topologically non-trivial gap. Note that the combined $(6\sqrt{3} \times 6\sqrt{3})$R30° super cell (Fig. 2e, f) leads to band backfolding which projects graphene bands (black) into the indenene gap. Importantly, in order to disentangle the role of $d_{In-Si}$ from the graphene-induced change in $\lambda_{ISB}$, we calculate both band structures at the In-Si bonding distance determined from XSW of intercalated indenene and for the latter with the corresponding indenene graphene distance. **e, f** Fourier filtered dI/dV maps of the same position, but taken at **e** −100 mV (VB) and **f** −250 mV (VB-1) reproducing the topologically non-trivial switch of the charge maximum from site B to A (arrow)[14] (details in Supplementary Note 8).

hydrogen and stabilizes the $(1 \times 1)$ SiC(0001) surface[30]. Employing a UHV suitcase, samples were subsequently transferred *in vacuo* to our UHV molecular beam epitaxy system, where the surface quality was confirmed by LEED.

Pristine 2 ML and 1 ML indium films were grown as we described elsewhere[18] by hydrogen desorption during In growth, while heating the samples to 600 °C for one minute under simultaneous indium (99.9999% purity) exposure from a Knudsen cell held at 770 °C. The substrate temperature was measured with a pyrometer (Keller, detection range 1.1–1.7 μm, emissivity $\epsilon = 85\%$) sensitive to a temperature range of 250–2000 °C. Subsequent to the H-desorption step, well-ordered films of reproducible quality and thickness were prepared by evaporating In onto a substrate held at 420 °C. This cycle is repeated eight times to produce In films of identical quality and thickness.

Intercalated 2 ML and 1 ML indium films were grown in two steps: First, we grow the $(6\sqrt{3} \times 6\sqrt{3})$R30° reconstruction, also referred to ZLG, by heating the substrate in Ar atmosphere (950 mbar) to 1360 °C for 15 min[31]. Second, we perform the intercalation by evaporating pure indium (≤99.9999%) from a Knudsen cell held on 775 °C in two steps[21]: (i) evaporating In for 10 min at RT and (ii) annealing the sample at 500 °C for 20 min. To yield optimized film quality, we repeat this process for five times followed by a post-annealing flash at 800 °C for 10 s.

Experiments on the stability of intercalated indenene at various conditions (discussed in Figs. 1d, e and 3b) were performed by

exposure to $O_2$ (21 kL, 99.9995% purity), air (10 min) and water (90s, CAS number 7732-18-5). Water exposure included additional air exposure for 5 min and was followed by a 90 min degas in ultra high vacuum <250 °C where no structural changes are expected for intercalated indenene.

After each process, samples were characterized by LEED. The LEED data shown in this work were taken at an electron kinetic energy of 100 eV.

## ARPES and XPS measurements

were performed in our home-lab setup equipped with a hemispherical analyzer (PHOIBOS 100), a He-VUV lamp (UVS 300: Fig. 1a, μSIRIUS: Fig. 1b, 21.2 eV), an umonochromatized Al K-$\alpha$ light source, and a 6-axis manipulator capable of LHe-cooling to 20 K. ARPES and XPS data shown in Fig. 1a (Figs. 1b, d, e and 3a, b) were recorded at 20 K (RT) and a base pressure of <10⁻¹⁰ mbar. Differential pumping of the He-VUV-lamp kept the base pressure below 10⁻⁹ mbar during the ARPES measurements. Further ARPES data were taken at the beamlines I05 of Diamond Light Source, England (Fig. 2b and Supplementary Fig. 4c, d) and MAESTRO of Advanced Light Source, USA (Fig. 4a,b). The measurement in Fig. 2b (Fig. 4a, b and Supplementary Fig. 7) was taken with linear horizontally polarized light with a photon energy of 46 eV (90 eV) at RT (260 K), while the pressure was ≤3 × 10⁻¹⁰ mbar. XPS data shown in Fig. 1d, e are corrected by subtraction of a Shirley background.

## STM measurements

STM data were acquired at 4.7 K and a base pressure lower than $5 \times 10^{-11}$ mbar (Omicron low-temperature LT STM) using a chemically etched W-tip that was characterized by imaging the Ag(111) surface state. Differential conductance (dI/dV) maps were taken at constant height using a standard lock-in technique with a modulation frequency of 971 Hz and modulation voltage of $V_{rms}$=15 mV. Point spectroscopy dI/dV curves were recorded using the same lock-in technique (Fig. 2g: intercalated indenene: $V_{rms}$=20 mV, pristine indenene: $V_{rms}$=10 mV). FFT-filtering (Hanning window function) of the dI/dV-maps shown in Fig. 4e, f was carried out using the software IGOR.

## Error analysis of In-Si bonding distance

The intrinsic precision is typically negligible compared to the variance of different XSW measurements[32]. Potential sources of the latter are small instabilities in beamline, manipulator, and analyzer. The error bar shown in Fig. 1c represents the standard error determined from two XSW datasets per datapoint.

## STEM measurements

Cross-sectional lamellae for scanning transmission electron microscopy (STEM) investigations were prepared on a Dual-Beam System (FEI Helios Nanolab). Lamella preparation starts with the beam-induced deposition of a Pt ridge to protect the surface. After Ga ion beam milling, the lamella is lifted out using an Omniprobe micro-manipulator and attached to a Cu transmission electron microscope (TEM) grid. The lamella is then thinned to electron transparency using several polishing steps first at 30 keV and finally at 2 keV ion energy. Transfer to the TEM was carried out ex situ, exposing the lamella to ambient air for several minutes.

STEM measurements were taken in an uncorrected FEI Titan 80-300 microscope operating at 300 kV acceleration voltage, a beam current of 100–120 pA, a convergence semiangle of 10 mrad and dwell time of 10-20 μs per pixel. The STEM image in Fig. 2 and Supplementary Note 2 were taken in high angle annular dark field (HAADF) mode (scattering angles between 40 and 250 mrad). The spatial resolution under this condition is on the order of 140 pm. A gamma filter with $g = 0.5$ was applied to the HAADF Signal in order to enhance the weak signal of the graphene layer.

## DFT calculations of indenene

presented in Fig. 1 are performed within the density functional theory framework as implemented in the Vienna ab initio simulation package (VASP) within the projector-augmented plane-wave (PAW) method[33,34]. For the exchange-correlation potential the HSE06 functional was used[35] by expanding the Kohn-Sham wave functions into plane-waves up to an energy cutoff of 500 eV. We sampled the Brillouin zone on an $12 \times 12 \times 1$ regular mesh with self-consistently included SOC[36]. We consider a $(1 \times 1)$ reconstruction of triangular In on four layers of Si-terminated SiC(0001) with an in-plane lattice constant of 3.07 Å. The equilibrium structure is obtained by relaxing all atoms until all forces converged below 0.005 eV/Å resulting in an In-SiC distance of $d_{In-Si} = 2.68$ Å. To disentangle the electronic states of both surfaces a vacuum distance of at least 25 Å between periodic replicas in z-direction is assumed and the dangling bonds of the substrate-terminated surface are saturated by hydrogen. For our study on intercalated indenene, we consider a monolayer SiC substrate with hydrogen-passivated dangling bonds on the unphysical surface. To compensate for the increased computational costs inherent to the $(6\sqrt{3} \times 6\sqrt{3})R30°$ super-cell reconstruction (depicted in Fig. 2e,f and containing 108 In and 338 graphene C atoms), DFT calculations on intercalated indenene are performed within the Perdew-Burke-Ernzerhof (PBE) scheme. Note that the choice of the exchange functional (HSE06 vs. PBE) has no influence in related studies[32] and no significant impact on the SOC and ISB induced band splitting at the K-points of pristine indenene[14] discussed in Fig. 4. Here, we used a plane-wave cutoff of 300 eV, together with a $12 \times 12 \times 1$ (Fig. 4c) and $2 \times 2 \times 1$ (Fig. 4d) $k$ mesh and SOC included self-consistently[36]. For the latter, we additionally consider van der Waals corrections according to ref. 37. The graphene buckling was studied by applying a selection of van der Waals corrections[32,37], altogether yielding a corrugation smaller than 0.01 Å, thus representing quasi-planar graphene.

## Data availability

The raw data generated in this study have been deposited in the WueData database[38] under accession code https://doi.org/10.58160/126.

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

## Acknowledgements

We are grateful for funding support from the Deutsche For-schungsgemeinschaft (DFG, German Research Foundation) under Ger-many's Excellence Strategy through the Würzburg-Dresden Cluster of Excellence on Complexity and Topology in Quantum Matter ct.qmat (EXC 2147, Project ID 390858490) as well as through the Collaborative Research Center SFB 1170 ToCoTronics (Project ID 258499086). We acknowledge Diamond Light Source for time on beamline I09 and I05 under proposals SI31808-1, SI25151-4, and SI30583-1. This research further used resources of the Advanced Light Source, which is a DOE Office of Science User Facility under contract no. DE-AC02-05CH11231. We gratefully acknowledge the Gauss Centre for Supercomputing e.V. (https://www.gauss-centre.eu) for funding this project by providing computing time on the GCS Supercomputer SuperMUC-NG at Leibniz Super-computing Centre (https://www.lrz.de).

## Author contributions

C.S. and J.E. have realized the epitaxial growth and surface character-ization and carried out the photoelectron and scanning tunneling spectroscopy experiments and their analysis. P.E. and S.E. have con-ceived the theoretical ideas and performed the DFT, Wannier and Ber-ryology calculations. On the experimental side, contributions came from M.S., K.L., P.K., T.W., Me.S., B.L., M.K., V.J., C.J., A.B., E.R., T.K., C.C., T.-L.L., S.M., and R.C., while G.S. gave inputs to the theoretical aspects. R.C. and S.M. supervised this joint project and wrote the manuscript together with all other authors.

## Funding

## Competing interests

The authors declare no competing interests.
