## [Peer Review File · Nature Communications]

REVIEWER COMMENTS

Reviewer #1 (Remarks to the Author):

This manuscript demonstrates the synthesis of monolayer indinene via intercalation under graphene on SiC. The goal is to develop an air stable, room temperature quantum spin hall insulator. The synthesis is similar to previous reports of “confinement heteroepitaxy” (N. Briggs, et. al. Nature Materials 19, 637-643 2020), in which simple metal films with a self-limited thickness of a few atomic layers grow via intercalation under graphene on SiC. The difference in the present manuscript is that the authors show that post growth annealing can decrease the indium thickness from 2 atomic layers to 1 atomic layers. Core level photoemission measurements confirm that the 1-layer indinene is protected from oxidation by the graphene, and the near EF bands measured by ARPES are generally consistent with bandstructure calculations that predict a QSHI gap. However, it should be noted that since the graphene covered structure is hole doped, the gap at the indinene K point, which is the hallmark of the QSHI state, is pushed above the Fermi energy and is not observable by ARPES. The authors rely on comparison between ARPES-measured filled states and bandstructure calculations to infer that the gap is likely preserved in the graphene-covered structure.

I agree that intercalation under graphene is a nice strategy to enable QSHI-based devices and ex situ measurements, and the work is important to be published somewhere. A direct measurement of the QSHI state in an intercalated sample is clearly missing. I understand this is difficult via ARPES, since the gap lies above the Fermi energy. Perhaps surface alkali doping may be sufficient to push the gap below EF for ARPES? Alternatively, since the samples are now protected by graphene, presumably a transport measurement could be performed that establishes the QSHI state. I realize these are potentially difficult experiments, but these are in my view the missing pieces that would make this manuscript excellent.

I have a few other questions and suggestions

1. An EDC linewidth analysis of pristine versus intercalated would help for quantifying the disorder induced by the graphene cap (fig 2).
2. Similarly, EDC analysis of the intercalated sample before and after exposure to water and annealing (fig 4). In Fig 4 I also suggest directly comparing the near EF features at indinene K before and after water exposure. This seems more relevant than examining graphene K after water (fig 4d).
3. When annealing to go from 2 layers to 1 layer indium, where does the indium go? Is it sublimating through openings in graphene? Or does it segregate? How applicable is this strategy for making intercalated monolayers of other materials, e.g. Ga?

Reviewer #2 (Remarks to the Author):

In this manuscript, the authors present a concept for stabilizing the already recently published QSHI phase of thin In-layers on SiC surfaces. The realization of monolayer structures on semiconducting surfaces is a powerful tool for the growth of novel quantum phases and has been impressively demonstrated in the past using superconducting and topologically non-trivial 2D materials as examples. Stabilization, i.e. essentially immunization against oxidation, is achieved by intercalation experiments on buffer layers on SiC(0001). After intercalation of In, the same In phases are formed at the interface, but this time protected by more or less free-standing graphene, as shown by (massive) oxidation experiments.

Such passivation strategies are indeed crucial for envisaged transport experiments, as is emphatically mentioned in the introduction. Unfortunately, such transport measurements could not be performed so far. At the end, the authors themselves point out the main problem that the delaminated graphene itself is conductive. Furthermore, it is not clear how large the In-domains are and what is the influence of the SiC substrate steps. On the other hand, the authors impressively show by means of photoelectron spectroscopy in detail that the (already known) QSHI phase is stabilized despite passivation by graphene and discuss in particular the influence of bond spacing and In- coverage. Against this background, the authors fall a little short of expectations and should perhaps tone down the introduction a bit. Besides, I have the following comments:

- The STEM resolution is quite poor (maybe only the reproduction of the image). It is not clear from the image, if the orientation of the lamella is correct or if a further SiC terrace is hiding an In layer. The In-In and SiC distances are very similar, so it is difficult to discriminate. By the way, did the authors measure also the $d_{\text{In-In}}$ distance with XSW for the 2 ML case?

- No evidence is provided regarding the homogeneity. It is not clear that the STEM and the (integrating) ARPES measurements were taken from the same areas. Since the group is well known for their STM capabilities, it would be fair to show large areas and high-resolution STM images to demonstrate the homogeneity and provide more evidence for 1 and 2 ML In phases at the interface.

- What causes the different n-type doping of graphene? As pointed out by the authors, also incomplete intercalation was suggested for the n-type doping of graphene (ref 28). Here, the authors use it as a benchmark to distinguish between 2ML and 1ML intercalation thicknesses. Therefore, the demonstration of homogeneity is even more important in order to allow for this statement.

Therefore, I cannot recommend publication of the work in Nat. Com.

Reviewer #3 (Remarks to the Author):

Although interesting, I don't think that this paper can be published in Nature Comm., as it looks to me rather a follow-up of previous works.

Besides, the content is not so different from the intercalation of Plumbene :

« Proximity-Induced Gap Opening by Twisted Plumbene in Epitaxial Graphene » by C. Ghosal et al., PRL 129, 116802 (2022) .

I think that the conductive nature of the graphene cap, honestly indicated by the authors lines 274-276, may cause a real problem.

More convincing would be the in-situ capping by h-BN, although this would not be easily scalable.

Finally, the authors are too negative about silicene, germanene and stanene (lines 54-59). Some papers, as below, appear encouraging :

Tsai et al., Phys. Chem. Chem. Phys., 2015, 17, 21389

Zhang et al., PRL, 116, 256804 (2016)

Zhuang et al., Adv. Sci., 2018, 5, 1800207

Stabilizing an atomically thin quantum spin Hall insulator at ambient conditions: Graphene-intercalation of indenene

- Response to Reviewers -

December 19, 2023

Reviewers' comments to the Authors:

Reviewer #1:

This manuscript demonstrates the synthesis of monolayer indenene [sic] via intercalation under graphene on SiC. The goal is to develop an air stable, room temperature quantum spin Hall insulator. The synthesis is similar to previous reports of "confinement heteroepitaxy" (N. Briggs, et. al. Nature Materials 19, 637-643 2020), in which simple metal films with a self-limited thickness of a few atomic layers grow via intercalation under graphene on SiC. The difference in the present manuscript is that the authors show that post growth annealing can decrease the indium thickness from 2 atomic layers to 1 atomic layers. Core level photoemission measurements confirm that the 1-layer indenene is protected from oxidation by the graphene, and the near EF bands measured by ARPES are generally consistent with band structure calculations that predict a QSHI gap. However, it should be noted that since the graphene covered structure is hole doped, the gap at the indenene K point, which is the hallmark of the QSHI state, is pushed above the Fermi energy and is not observable by ARPES. The authors rely on comparison between ARPES-measured filled states and band structure calculations to infer that the gap is likely preserved in the graphene-covered structure. I agree that intercalation under graphene is a nice strategy to enable QSHI-based devices and *ex situ* measurements, and the work is important to be published somewhere. A direct measurement of the QSHI state in an intercalated sample is clearly missing. I understand this is difficult via ARPES, since the gap lies above the Fermi energy. Perhaps surface alkali doping may be sufficient to push the gap below EF for ARPES?

Response:

We appreciate the reviewers' interest in our work and his/her critical remarks which - as detailed further below - has helped us to improve the paper considerably. We also wish to emphasize that our current results are not just merely "similar to previous reports" on the synthesis of intercalated few-layer In films. In contrast to them, we provide here a detailed comparison of pristine and graphene-intercalated indenene monolayers, presenting unambiguous experimental proof that the electronic structure of the intercalated film remains unaffected by the graphene capping (apart from a small charge transfer). In fact, with new STM data amended to our manuscript we are even able to demonstrate that the topologically non-trivial nature of the intercalated indenene is preserved (see below), which removes the reviewer's objection about a lack of QSHI evidence.

Furthermore, our study demonstrates that already a monolayer of graphene is sufficient to protect the indenene film from oxidation and/or contamination when exposed to ambient atmosphere or even immersed in liquid water. This opens unprecedented opportunities for *ex situ* experiments on these otherwise highly delicate atomic In monolayers. While, admittedly, graphene due its own conductive properties may not be the ideal passivation material, our study nonetheless carries the important message that van der Waals capping is an amenable route to allow practical device fabrication from (topological) atomic monolayers. Future studies will look at the use of, e.g., hBN which however requires a different deposition technique. In any case, our present results go clearly far beyond the existing literature.

Concerning the energy position of the band gap, we wish to point out that this can be controlled by substrate doping concentration and/or alkali atom adsorption as correctly remarked by the reviewer. Most of the data shown in our manuscript are taken on intercalated indenene grown on higher n-doped SiC substrates, pushing the Fermi level *inside* the global (non-trivial) band gap – a situation one would favor for transport experiments. Making the band gap fully accessible to ARPES (i.e., by pushing the conduction band minimum below E_F) can indeed be achieved by further n-doping via alkali metal deposition. We used potassium doping for this purpose and now show the resulting K-point ARPES spectra in Fig. 4(a,b) of our manuscript and Fig. S7 in Supplementary Note 6.

In the following we address the additional questions raised by Reviewer #1.

Comment "0":

Alternatively, since the samples are now protected by graphene, presumably a transport measurement could be performed that establishes the QSHI state. I realize these are potentially difficult experiments, but these are in my view the missing pieces that would make this manuscript excellent.

Response:

We agree with the reviewer that a direct confirmation of the QSHI phase by edge transport would be desirable. However, due to the conducting nature of the graphene overlayer such experiments would be extremely challenging, as they would require an unambiguous decomposition of indenene and graphene contributions to the measured current.

However, we can offer here two alternative experimental indicators of non-trivial topological behavior. The first is of indirect nature and concerns the In-Si bond length as determined by XSW photoemission. Our data clearly show that it falls into the topological sector of the phase diagram. A more direct proof is related to the energy sequence of the orbital angular momentum (OAM) eigenstates defining the band gap. As already shown in Ref.[1] for pristine indenene, the topological phase is characterized by an alternating stacking of the OAM states. Via wavefunction interference (see Ref. [1]) this translates into alternating charge localization in the unit cell, directly accessible to LDOS measurements by STM/STS. We observe exactly this behavior also for the intercalated indenene, thus providing unambiguous experimental proof of its topological character. This new data are now included in Fig. 4e,f and also in the Supplementary Note 8 (Fig. S9).

Comment 1:

I have a few other questions and suggestions:

1. An EDC linewidth analysis of pristine versus intercalated would help for quantifying the disorder induced by the graphene cap (fig 2).

Response:

We thank the reviewer for this useful suggestion and include an EDC linewidth analysis for pristine and intercalated indenene as well as after exposure to water in the Supplementary Note 5 (Fig. S5). Note, that the pristine indenene data are shifted in energy to align them with the p-doped intercalated indenene. All three EDCs display the same linewidth, clearly signaling that the graphene capping induces no additional disorder, not even after immersion in water. This finding is corroborated by the large-scale STM images in Fig. 2d and in the Supplementary Note 3.

Comment 2:

2. Similarly, EDC analysis of the intercalated sample before and after exposure to water and annealing (fig 4). In Fig 4 I also suggest directly comparing the near EF features at indenene K before and after water exposure. This seems more relevant than examining graphene K after water (fig 4d).

Response: As suggested by the reviewer, in the previous point we also added an EDC linewidth analysis before and after water exposure. Indeed, the indenene K-point after water is more relevant than the graphene K-point and was added in the Supplementary Note 5 (Fig. S5). It is complemented by a direct comparison of bandmaps before and after water exposure in Fig. S6 (Supplementary Note 5).

Comment 3:

3. When annealing to go from 2 layers to 1 layer indium, where does the indium go? Is it sublimating through openings in graphene? Or does it segregate? How applicable is this strategy for making intercalated monolayers of other materials, e.g. Ga?

Response:

We thank the referee for this interesting question. The temperature (550°C) of this annealing step is little higher than the intercalation temperature and thus principally allows the indium atoms to pass through the graphene barrier in terms of an inverse intercalation process. As correctly indicated by the referee, defects or openings in the graphene lattice might play a significant role in this mechanism similar to intercalation itself [2, 3]. This obviously requires breaking of the interlayer In-In bonds as well as a high mobility of the In atoms to reach these lattice faults. Both conditions are fulfilled at $\approx 480^\circ\text{C}$ for the related In/SiC system without graphene capping [4] or In/Si(111) [5]. Beneath graphene, indium can be assumed similarly mobile as demonstrated in a very recent study that applied voltage pulses with STM tips to move larger indium amounts (of 2 ML/SiC) on the nm-scale [6]. The confinement by the graphene overlayer is most likely responsible for the slightly higher temperature (by 70°C) necessary for this annealing step which might require even higher temperatures for thicker graphene cappings. After reaching the other side of graphene, the metal has to leave the surface or segregate out of the region of interest. The former case is facilitated by the high vapour pressure of indium [7] at this temperature and ultrahigh vacuum conditions.

This assessment can be expanded to deintercalation of other materials from the graphene/SiC interface. For the particular case of gallium mentioned by the referee a single layer is predicted to be stable, while experimentally the intercalation of 2 ML has been demonstrated [8]. However, the lower vapour pressure might pose problems in the desorption step after deintercalation [7]. Instead of long time annealing, we speculate that a short (10 s) temperature flash (1000°C) might suffice to desorb deintercalated Ga without destroying the underlying structure. While a better understanding of the detailed (de)intercalation processes is clearly desired, it is not at the specific focus of our present study and must be left to future studies.

In this context, we would like to add that during revision of the manuscript we further developed our recipe for In intercalation to achieve even better film quality. We now grow the zero-layer graphene in 950 mbar Ar atmosphere, as reported in [9, 10]. An additional post-growth flash at 800°C ensures large-scale defect-free indenene intercalation. This new recipe has allowed us to reproduce all measurements with considerably enhanced sample quality.

We hope that reviewer #1 finds our above response satisfactory and the revised version of our manuscript now suitable for acceptance.

Reviewer #2:

In this manuscript, the authors present a concept for stabilizing the already recently published QSHI phase of thin In-layers on SiC surfaces. The realization of monolayer structures on semiconducting surfaces is a powerful tool for the growth of novel quantum phases and has been impressively demonstrated in the past using superconducting and topologically non-trivial 2D materials as examples. Stabilization, i.e. essentially immunization against oxidation, is achieved by intercalation experiments on buffer layers on SiC(0001). After intercalation of In, the same In phases are formed at the interface, but this time protected by more or less free-standing graphene, as shown by (massive) oxidation experiments. Such passivation strategies are indeed crucial for envisaged transport experiments, as is emphatically mentioned in the introduction. Unfortunately, such transport measurements could not be performed so far. At the end, the authors themselves point out the main problem that the delaminated graphene itself is conductive. Furthermore, it is not clear how large the In-domains are and what is the influence of the SiC substrate steps. On the other hand, the authors impressively show by means of photoelectron spectroscopy in detail that the (already known) QSHI phase is stabilized despite passivation by graphene and discuss in particular the influence of bond spacing and In-coverage. Against this background, the authors fall a little short of expectations and should perhaps tone down the introduction a bit.

Response:

We thank the reviewer for the appreciation of our work and his/her insightful comments. As correctly stated by the reviewer, the main message of our paper is the experimental verification that (1) the electronic and, in particular, topological properties of indenene do not get altered by graphene intercalation, and that (2) a single graphene layer suffices to protect the indenene monolayer against ambient conditions. This is the first step in demonstrating that van der Waals capping is a viable route to bring otherwise highly-delicate atomic monolayer systems into the realm of device physics while preserving their specific properties as ultimate 2D quantum materials.

While we partly understand the reviewer's disappointment that we do not present actual transport experiments, we have to point out that we are experts for surface science techniques such as ARPES and STM/STS, and not for quantum transport. This, however, does not prevent us from probing the topology of our intercalated indenene layers – we now have added new STM/STS data providing unambiguous and unique experimental evidence for the topological non-trivial nature of our intercalated indenene layers. This is not to say that experiments on edge transport are completely impossible, especially as the large-scale quality of our films and the terrace widths of the SiC substrate (up to 0.5 microns) are within the scope of modern nanodevice fabrication methods. They are just very challenging as they require an unambiguous decomposition of indenene and graphene contributions to the measured current. Such experiments are left for future work.

In the following we address the additional questions raised by Reviewer #2.

Comment 1:

Besides, I have the following comments:

- The STEM resolution is quite poor (maybe only the reproduction of the image). It is not clear from the image, if the orientation of the lamella is correct or if a further SiC terrace is hiding [sic] an In layer. The In-In and SiC distances are very similar, so it is difficult to discriminate. By the way, did the authors measure also the d_{In-In} distance with XSW for the 2 ML case?

Response:

We thank the referee for pointing out the limited quality of Fig. 2. In the meantime we have been able to obtain much better resolved STEM data for 1 ML and a 2 ML indium intercalated samples, respectively, now shown in Fig. 2 and Supplementary Note 2. Misaligned surface steps in the lamella indeed pose a risk of false interpretation. We thus aligned the mostly parallel surface steps along the electron beam direction during preparation of the lamella. A representative STEM image in the Supplementary Note 2 Fig. S2a confirms this orientation and demonstrates an indium coverage of 2 ML on each terrace. This is reproduced in further STEM images taken at individual terraces (Fig. S2b) that consistently show the same positions of the indium rows as a related study on 2 ML intercalated indium [8].

XSW measurements were only performed on the indenene monolayer samples. The thus determined

In-Si bond length places indenene clearly in the QSHI sector of the topological phase diagram.

Comment 2:

- No evidence is provided regarding the homogeneity. It is not clear that the STEM and the (integrating) ARPES measurements were taken from the same areas. Since the group is well known for their STM capabilities, it would be fair to show large areas and high-resolution STM images to demonstrate the homogeneity and provide more evidence for 1 and 2 ML In phases at the interface.

Response:

We are thankful for this comment. Further high resolution STM topography images were performed to demonstrate a micrometer scale single domain growth of the intercalated indenene and are added to Fig. 2d and Supplementary Note 3 (Fig. S3). Due to this large area growth, the characterization of samples by ARPES and STM can be easily transferred to STEM. In addition to that, high-resolution STM images for both, indenene and graphene, are implemented in the new Figs. 2(e),(f), respectively. Depending on bias voltage we can tunnel specifically into indenene or graphene states of the very same region, thereby allowing us to map both monolayers separately.

Comment 3:

- What causes the different n-type doping of graphene? As pointed out by the authors, also incomplete intercalation was suggested for the n-type doping of graphene (ref 28). Here, the authors use it as a benchmark to distinguish between 2ML and 1ML intercalation thicknesses. Therefore, the demonstration of homogeneity is even more important in order to allow for this statement. Therefore, I cannot recommend publication of the work in Nat. Com.

Response:

This is indeed an interesting question. As shown in Tab. 1 of the Supplementary Note 4, we observe an decreasing n-doping of the graphene overlayer after deintercalation of the second In layer. Related studies on epitaxial and quasi free-standing graphene on SiC(0001) thoroughly discussed in detail the charge transfer from the ZLG [11], the spontaneous polarization of the polar 4H-SiC [12, 13] and the space charge region formed in SiC [12] as most relevant doping sources of graphene on SiC. These mechanisms also need to be carefully considered for intercalated indium and are certainly weighted differently for bi- and single-layer indium causing the observed graphene band fillings. However, a detailed analysis of the microscopic doping mechanisms and their dependence on indium coverage is well beyond the scope of the present study and instead requires a dedicated SiC polytype-dependent study as demonstrated in Ref. [12] as well as adequate modelling.

Based on (i) the large-scale homogeneity of our samples (see response to Comment 2), (ii) the unambiguous agreement between the ARPES spectra of the intercalated and pristine In films [1, 4], and (iii) the clear evidence from the STEM data, there is no doubt on the identification of bi- vs. monolayer In films.

We hope that reviewer #2 finds our above response convincing and the revised version of our manuscript now suitable for acceptance.

Reviewer #3:

Comment 1:

Although interesting, I don't think that this paper can be published in Nature Comm., as it looks to me rather a follow-up of previous works. Besides, the content is not so different from the intercalation of Plumbene : « Proximity-Induced Gap Opening by Twisted Plumbene in Epitaxial Graphene » by C. Ghosal et al., PRL 129, 116802 (2022) .

Response:

We appreciate that the reviewer finds our work interesting, however, vehemently disagree with his/her opinion that we contribute a follow-up of previous work. Our study demonstrates the ex-situ stabilization of the QSHI indenene and thereby overcomes a major obstacle for this material that almost all 2D atomic monolayer systems are facing, QSHIs in particular. We provide here a detailed comparison of pristine and graphene-intercalated indenene monolayers, presenting unambiguous experimental proof that the electronic structure of the intercalated film remains unaffected by the graphene capping (apart from a small charge transfer). In fact, with new STM data amended to our manuscript we are even able to demonstrate that the topologically non-trivial nature of the intercalated indenene is preserved. This goes far beyond the studies quoted by the reviewer and or any other existing literature.

Capping by graphene is obviously inspired by the wide field of intercalation that we acknowledge in the manuscript by citing studies relevant to our work. They include the interesting STM study on intercalated Pb mentioned by the referee as well as earlier reports on indium intercalation, which unfortunately not conclude on a precise structural model of the monolayer In. In order to demonstrate the indenene structure and non-trivial topology according to the high quality standards of Nature Communications, we complete this missing structural piece and scrutinize electronic and topological properties by a variety of spectroscopic techniques. Part of that is the hitherto first side-by-side comparison of an intercalated and uncapped monolayer/SiC material bearing particular relevance for related intercalation studies. Our results demonstrate the non-trivial topology of intercalated indenene fabricated by a very simple technique and thus deserve in our opinion publication in Nature Communications.

Comment 2:

I think that the conductive nature of the graphene cap, honestly indicated by the authors lines 274-276, may cause a real problem. More convincing would be the in-situ capping by h-BN, although this would not be easily scalable.

Response:

We certainly agree with the reviewer that graphene is not the most ideal capping material when targeting at, e.g., edge transport measurements. This is not to say that such experiments are completely impossible, especially as the large-scale quality of our films and the terrace widths of the SiC substrate (up to 0.5 microns) are within the scope of modern nanodevice fabrication methods. They are just very challenging as they require an unambiguous decomposition of indenene and graphene contributions to the measured current.

In contrast, hBN may indeed seem a better choice for this particular purpose. At the same time, deposition of an hBN capping layer is far more complicated than the synthesis of graphene mono- and multilayers from a SiC substrate (which in our case comes for free!) and indeed not as easily scalable, as correctly pointed out by the reviewer.

Therefore, our current study on graphene intercalation has to be viewed as a first step in demonstrating that van der Waals capping is a viable route to bring otherwise highly-delicate atomic monolayer systems into the realm of device physics while preserving their specific properties as ultimate 2D quantum materials. In fact, our experimental proof of non-trivial topology in the intercalated indenene monolayer is the first of its kind in any intercalation study so far.

Comment 3:

Finally, the authors are too negative about silicene, germanene and stanene (lines 54-59). Some papers, as below, appear encouraging: Tsai et al., Phys. Chem. Chem. Phys., 2015, 17, 21389 Zhang et al., PRL, 116, 256804 (2016) Zhuang et al., Adv. Sci., 2018, 5, 1800207

Response:

We thank the reviewer for pointing out interesting work on germanene synthesis. These studies nicely show germanene growth on various semiconducting substrates in the 2 nm to monoatomic thickness range, yet despite the demonstration of Dirac states, the characterization of the electronic properties lack proof of an inverted, topologically non-trivial band gap. Our statement in lines 54-59 emphasizes the need to demonstrate both topological non-triviality and growth on a semiconducting substrate, which unfortunately is not yet met by group IV-based QSHI monolayers. To make this even clearer we have rephrased this sentence now in the following form:

But while spin-orbit coupling (SOC) in graphene is too weak to open an appreciable band gap, monolayers built from heavier group IV elements such as silicene, germanene, and stanene [14, 15] failed to display a large non-trivial band gap when placed on a semiconducting substrate [16].

We hope that reviewer #3 finds our reasoning sufficiently convincing to reconsider his/her opinion.

References

- [1] Bauernfeind, M. *et al.* Design and realization of topological Dirac fermions on a triangular lattice. *Nat. Commun.* **12**, 5396 (2021).
- [2] Liu, Y. *et al.* Mechanism of metal intercalation under graphene through small vacancy defects. *J. Phys. Chem. C* **125**, 6954–6962 (2021).
- [3] Lu, X., Liu, Y., Shao, M. & Liu, X. Defect-mediated intercalation of dysprosium on buffer layer graphene supported by SiC(0001) substrate. *Chem. Phys. Lett.* **742**, 137162 (2020).
- [4] Erhardt, J. *et al.* Indium epitaxy on sic(0001): A roadmap to large scale growth of the quantum spin Hall insulator indenene. *J. Phys. Chem. C* **126**, 16289–16296 (2022).
- [5] Surnev, S. L., Kraft, J. & Netzer, F. P. Modification of overlayer growth kinetics by surface interlayers: The Si(111) $\sqrt{7} \times \sqrt{3}$ -indium surface. *J. Vac. Sci. Technol. A* **13**, 1389–1395 (1995).
- [6] Pham, V. D., Dong, C. & Robinson, J. A. Atomic structures and interfacial engineering of ultrathin indium intercalated between graphene and a SiC substrate. *Nanoscale Adv.* **5**, 5601–5612 (2023).
- [7] Honig, R. E. Vapor pressure data for solid and liquid elements. *RCA Review* **23**, 567 (1962).
- [8] Briggs, N. *et al.* Atomically thin half-van der Waals metals enabled by confinement heteroepitaxy. *Nat. Mater.* **19**, 637–643 (2020).
- [9] Emtsev, K. *et al.* Towards wafer-size graphene layers by atmospheric pressure graphitization of silicon carbide. *Nat. Mater.* **8**, 203–207 (2009).
- [10] Ostler, M., Speck, F., Gick, M. & Seyller, T. Automated preparation of high-quality epitaxial graphene on 6H-SiC(0001). *Phys. Status Solidi B* **247**, 2924–2926 (2010).
- [11] Kopylov, S., Tzalenchuk, A., Kubatkin, S. & Fal’ko, V. I. Charge transfer between epitaxial graphene and silicon carbide. *Appl. Phys. Lett.* **97**, 112109 (2010).
- [12] Mammadov, S. *et al.* Polarization doping of graphene on silicon carbide. *2D Mater.* **1**, 035003 (2014).
- [13] Ristein, J., Mammadov, S. & Seyller, T. Origin of doping in quasi-free-standing graphene on silicon carbide. *Phys. Rev. Lett.* **108**, 246104 (2012).
- [14] Liu, C.-C., Feng, W. & Yao, Y. Quantum spin Hall effect in silicene and two-dimensional germanium. *Phys. Rev. Lett.* **107**, 076802 (2011).
- [15] Xu, Y. *et al.* Large-Gap Quantum Spin Hall Insulators in Tin Films. *Phys. Rev. Lett.* **111** (2013).
- [16] Kou, L., Ma, Y., Sun, Z., Heine, T. & Chen, C. Two-Dimensional Topological Insulators: Progress and Prospects. *J. Phys. Chem. Lett.* **8**, 1905–1919 (2017).

REVIEWERS' COMMENTS

Reviewer #1 (Remarks to the Author):

I think the authors have sufficiently addressed all reviewer concerns. I recommend publication after a minor clarification: In figure 4(f), I think the color scale should be labelled the same as in panel (e), i.e. should be "FFT filtered dI/dV ." Additionally it appears that they use the same black-blue-yellow color table as in (e), and thus the color intensity bar should be updated accordingly. Or does this white-blue-black color table correspond to the FFT inserts?

Reviewer #2 (Remarks to the Author):

Thank you for sending me the revised version.

The authors have provided answers to most of my questions and comments and have also presented some new data. In connection with the STM data, I noticed that the large-scale images in the supplement also show $\sqrt{3}$ reconstructed graphene areas. The STEM images are now more convincing. Nevertheless, I am somewhat surprised that no knock-on damages occurred at 300kV, or that the barely recognizable graphene structure (independent of the detector) could possibly be a sign of this damage?

Unfortunately, the authors have not complied with my request to present the manuscript in a less sensational way in terms of electronic transport. The abstract promises great potential here, although there isn't any reliable transport data and in the end the authors themselves see problems with vdW capping.

All in all, the authors have greatly revised the manuscript and now show that the QSH phase is protected against water, etc. by the graphene. Unfortunately, the new physical aspects of the In-bilayer and the different doping behavior outlined more in the first draft have now been moved to the supplement. In my opinion, this has brought the revised manuscript focusing on the monolayer phase closer to previously published data. I believe that the manuscript does not fulfill the criteria of Nat. Comm. and can therefore unfortunately not propose it for publication.

Reviewer #3 (Remarks to the Author):

As authors, you have done a good job upon strongly improving the manuscript with new key elements. Besides, when necessary, your rebuttals are sound and convincing. Hence, I recommend the revised manuscript for publication as it is now.

Achieving environmental stability in an atomically thin quantum spin Hall insulator via graphene intercalation

- Response to Reviewers -

January 15, 2024

Reviewers' comments to the Authors:

Reviewer #1:

I think the authors have sufficiently addressed all reviewer concerns. I recommend publication after a minor clarification: In figure 4(f), I think the color scale should be labelled the same as in panel (e), i.e. should be "FFT filtered dI/dV." Additionally it appears that they use the same black-blue-yellow color table as in (e), and thus the color intensity bar should be updated accordingly. Or does this white-blue-black color table correspond to the FFT inserts?

Response:

We thank the reviewer for the recommendation to publish our work. For clarity, we moved the FFT magnitude spectra formally shown in the insets Fig. 4e,f and the corresponding color bar to Supplementary Note 8.

Reviewer #2:

Thank you for sending me the revised version. The authors have provided answers to most of my questions and comments and have also presented some new data. In connection with the STM data, I noticed that the large-scale images in the supplement also show sqrt3 reconstructed graphene areas.

Comment 1: The STEM images are now more convincing. Nevertheless, I am somewhat surprised that no knock-on damages occurred at 300kV, or that the barely recognizable graphene structure (independent of the detector) could possibly be a sign of this damage?

Comment 2: Unfortunately, the authors have not complied with my request to present the manuscript in a less sensational way in terms of electronic transport. The abstract promises great potential here, although there isn't any reliable transport data and in the end the authors themselves see problems with vdW capping.

All in all, the authors have greatly revised the manuscript and now show that the QSH phase is protected against water, etc. by the graphene. Unfortunately, the new physical aspects of the In-bilayer and the different doping behavior outlined more in the first draft have now been moved to the supplement. In my opinion, this has brought the revised manuscript focusing on the monolayer phase closer to previously published data. I believe that the manuscript does not fulfill the criteria of Nat. Comm. and can therefore unfortunately not propose it for publication.

Response to Comment 1:

We appreciate that the reviewer finds our updated STEM analysis of the indium coverage convincing. As shown in Ref. [1], knock-on damage of graphene is negligible at 300 kV, if the electron dose is kept low. This can be achieved by short dwell times (5-30 μ s) when rastering the beam [1]. For clarity we added the dwell time used (10-20 μ s) in our measurements to the Method section. Similarly, beam damage of indenene is only observed for very long dwell times (100 ms - 1 s).

As stated in Supplementary Note 2, the weak graphene contrast occurs due to the Z -dependence inherent to STEM. The presence of graphene is nevertheless evident in the horizontally integrated profiles shown red in Fig. 2c and Supplementary Fig. 2. Additionally, the graphene carbon rows are not resolved due to the limited experimental resolution as well as their orientation in the lamella.

In this regard, we note that the graphene layer was intact before the STEM measurement, as concluded from the indium adsorption position directly above the top Si-rows. In case of significantly damaged graphene, indium would oxidize during the *ex situ* preparation step and shift its position between those Si-rows [2].

Response to Comment 2:

Exploiting the edge channels of QSHIs is obviously the long-term motivation of our work and therefore must be mentioned in abstract. Albeit these experiments are challenging *directly at* graphene-intercalated indenene, as we repeatedly mention in the manuscript, our work contributes a crucial step to this objective, (1) by showing that van der Waals capping does not interfere with the topological properties of indenene and (2) by providing a calibration system for the assessment of *future* transport-suited capping materials, such as e.g. hBN.

Nonetheless, we are willing to tune down the formulation in the abstract by accepting the suggested version of the editor.

Reviewer #3:

Comment 1:

As authors, you have done a good job upon strongly improving the manuscript with new key elements. Besides, when necessary, your rebuttals are sound and convincing. Hence, I recommend the revised manuscript for publication as it is now.

Response:

We thank Reviewer 3 for acknowledging the improvements to the manuscript and recommending publication.

References

- [1] Xu, Q. *et al.* Controllable atomic scale patterning of freestanding monolayer graphene at elevated temperature. *ACS Nano* **7**, 1566–1572 (2013).
- [2] Erhardt, J. *et al.* Indium epitaxy on sic(0001): A roadmap to large scale growth of the quantum spin Hall insulator indenene. *J. Phys. Chem. C* **126**, 16289–16296 (2022).